# Enhanced XRF Methods for Investigating the Erosion-Resistant Functional Coatings

**Mihail Lungu [1], Cosmin Dobrea [1,2] and Ion Tiseanu [1,\*]**

[1]  National Institute for Laser, Plasma and Radiation Physics (NILPRP), 077125 Bucharest, Romania; lungumihail1989@yahoo.com (M.L.); cosmin.dobrea@inflpr.ro (C.D.)

[2]  Doctoral School of Science and Material Engineering, Technical University of Cluj-Napoca, 400114 Cluj-Napoca, Romania

\*  Correspondence: ion.tiseanu@inflpr.ro; Tel.: +4-021-457-4051

**Abstract:** The development of erosion-resistant functional materials usable as plasma facing first wall components (PFC) is crucial for increasing the lifetime of future fusion reactors. Generally, PFCs have to be quality checked and characterized regarding their composition, before integrating them into the fusion reactor vessel. Enhanced X-ray fluorescence (XRF) methods represent an effective alternative to conventional analysis methods for the characterization of refractive metallic coatings on large areas of fusion materials. We have developed and applied XRF methods as fast and robust methods for the characterization of the thickness and composition uniformity of complex functional coatings. These coatings consist of tungsten included in multilayer configuration and deposited on low or high Z substrates. We have further developed customized calibration protocols for quantifying the element composition and layer thickness of each investigated sample. The calibration protocols are based on a combination of standard samples measurements, Monte Carlo simulations, and fundamental parameter theoretical calculations. The calibrated results are discussed considering a selection of relevant PFC samples. The deposition uniformity was successfully investigated for different PFC-relevant tiles and lamella shaped samples with W layers below and over the W L-line saturation thickness. Also, the 2D thickness mapping capability of the XRF method was demonstrated by studying the plasma post-erosion pattern.

**Keywords:** X-Ray fluorescence; K-line; fundamental parameter; Monte Carlo simulations; multi-layer; thin films

## 1. Introduction

It is widely known that tungsten (W) is well suited as a possible candidate for plasma facing materials, because of its many properties, such as its low retention of reactor fuel, high melting point, and low sputtering yield [1]. During the last decade of fusion research, extensive progress was achieved in the development, production, and characterization of fusion relevant protective surface coatings that integrate W, designed to work in extreme conditions as components of tokamak first wall [2]. In contrast to a destructive analysis approach, non-destructive techniques such as X-ray fluorescence are more effective when investigating the plasma facing component (PFC) relevant samples. Such methods are applied as quality assurance tools, resulting in future improvements of layer deposition techniques [3,4]. More than that, the XRF method is considered an important tool with applicability in domains such as the environment [5], food [6], and metallurgy [7]. In general, the key advantages of XRF-based methods are expressed as follows: fast measuring, require no consumables, and are highly versatile [5].

Currently, the stoichiometry of PFCs is still not well-defined, as PFCs have a complex composition integrating numerous elements besides W [8]. During the last decade, different PFC relevant sample

candidates were investigated in the X-ray micro tomography laboratory. Their variation in stoichiometry, configuration, and substrate matrix determined the reconfiguration of the XRF experimental geometry, followed by the definition of several complex calibration protocols [9].

According to the literature, the XRF-based methods were successfully applied for the determination of elements with high atomic numbers (Z > 11), which are integrated in the reactor chamber [1]. More specifically, XRF methods were applied on samples retrieved from the fusion reactor to have an overview regarding the dust distribution and thin-layer integrity. This was conducted by analyzing the absorption of the X-ray lines emitted from the substrate [10], by the photon intensity calibration of each element, and by studying the intensity ratios between the investigated co-deposited layers [11]. The XRF method can be applied on large area mappings, thus providing a better understanding regarding the modifications occurring on PFC surfaces that can cause a negative impact on the thermos-mechanical resistance and fusion fuel retention properties [12].

In the current work, we considered the intensity variations of the fluorescence lines emitted from the W layer and Mo interlayer in relation to the substrate-based matrix effects. While integrating low and high energy X-ray sources (from 20 to >120 keV), the applicability of the XRF method was expanded for analyzing multilayers of W on Mo while integrating high and low Z substrates (W, C, and fine grain graphite (FGG)). These XRF methods were developed using the Tomo-Analytic instrument and the X-ray micro-tomography facility which were presented in-detail in previous articles [13,14]. To determine quantitative results, calibration procedures involving dedicated software, simulation algorithms and standard samples were applied.

## 2. Methods and Materials

### 2.1. X-Ray Fluorescence-Based Methods

The methods developed in the X-ray Microtomography Laboratory are the Low Energy X-ray fluorescence (micro-XRF/macro-XRF) and the High Energy X-ray fluorescence (W K-line XRF). These methods were described in previous papers [12] and proved their applicability in analyzing multiple co-deposited layers relevant for the fusion research domain.

The in house built Tomo-Analytic instrument integrates the micro-XRF/macro-XRF methods that are appropriate for investigating thin deposited layers commonly integrated in erosion witness markers which are commonly applied in fusion reactors. The experimental configuration of the Tomo-Analytic instrument includes components as: X-ray source with energies up to 50 keV (W, Ag, or Mo target), Si-pin energy selective detector and a high precision XYZ motorized manipulator used for profile and mapping measurements. A collimation system consisting of a polycapillary lens or a pin-hole pair of Pb disk can be fitted, depending on the required measuring resolution. The resolution is given by the focal spot size, which can range from tens to hundreds of micrometers on the surface of the investigated sample.

In comparison, the W K-line XRF method relies on an incident excitation beam with energies of 120–150 keV, appropriate for the excitation of more productive energetic K-lines of high Z elements (e.g. W). Excitation with higher photon energies extends the method's capability to be used for analyzing thicker samples. Therefore, the W K-line XRF method is able to characterize layer thicknesses over the saturation thickness of fluorescence integrated lines analyzed with the low energy XRF method.

An ad-hoc W K-line XRF setup was created within our in-house built X-ray micro-tomography facility. The experimental arrangement consists of a transmission X-ray source, with different anode targets such as W or Cu on diamond window. The experimental configuration of W K-line XRF implies the use of brass and lead collimators, hence producing a millimeter focal beam spot and also a Cd–Te X-ray detector that is designed for acquiring high energy fluorescence spectra.

As an example, by means of the W K-line XRF method, one can conduct experimental studies on the K characteristic energetic lines for W ($K_\alpha$ ~57–59 keV and $K_\beta$ ~67 keV). The energetic K-lines are more relevant because when comparing to the L-lines, they are less auto-absorbed in the investigated

deposited layer volume. This considerably increases the saturation threshold for the fluorescence signal acquired on a sample, and improves the overall precision and accuracy of the method [14].

The presented XRF-based methods rely on several calibration procedures in order to provide proper quantitative and qualitative results. In the following section, the most commonly applied protocols are exemplified and detailed.

## 2.2. Enhancing Calibration Protocols

It is well known that the non-destructive and fast XRF tools, which are widespread used in numerous research domains, relies extensively on calibration procedures in order to provide reliable quantitative and qualitative results on multilayer coatings or alloys [15–17]. A proper calibration consists of both measurements on standard samples and standardless procedures, like the Monte Carlo simulations and fundamental parameter (FP) theoretical calculations. Prior to determining the calibrated result, the standardless procedures could be assisted using experimentally determined calibration curves, if standard samples were available. Several protocols were applied in order to generate reliable results, as can be observed in the diagram below (Figure 1).

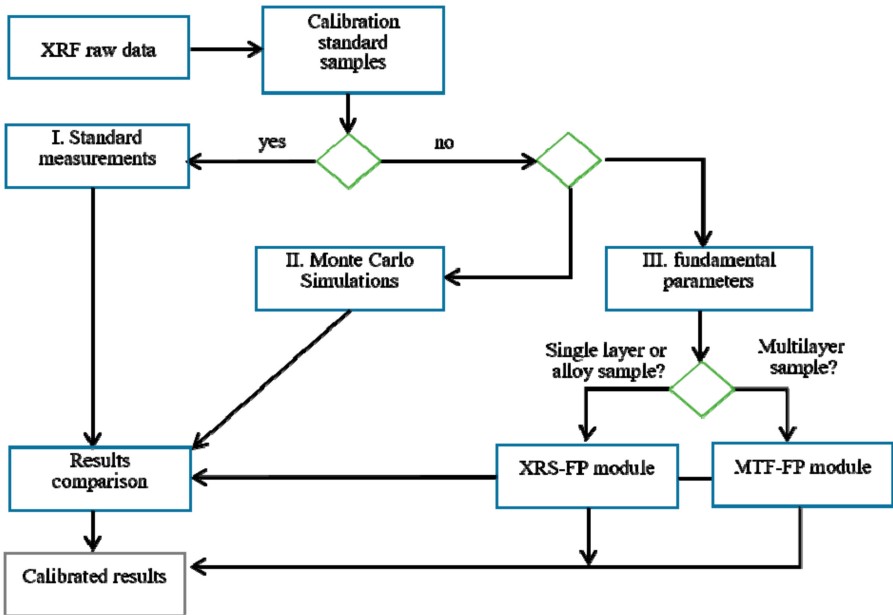

**Figure 1.** Calibration protocol for: standards measurements, Monte Carlo simulations, and fundamental parameters (FP) approach. MTF-FP—multi-layer analysis of thickness and composition.

The first protocol implies that the element dedicated calibration curves are generated based on the evaluation of standard/reference samples [18]. An energetic calibration implies the conversion from detector channels to energy values, while a quantitative calibration follows the transformation of the integrated peak of interest from the acquired spectra into the thickness/concentration of the investigated material. Therefore, the energetic and thickness calibration curves determined from the bulk materials and standard samples were applied to fluorescence acquired to raw spectra that is expressed in arbitrary units. The determination of the coating thickness resulting from converting the X-ray spectra intensity is highly dependent on the standard sample quality. Taking into account that the substrates, morphology, and matrix of the standard samples could limit the quality of the determined calibration curve, two distinct protocols were used. In the following, the Monte Carlo simulations and fundamental parameter approach are described.

The second protocol relies on Monte Carlo (MC)-based numerical simulations that represent an alternative to the standard sample-based calibration procedure, hence mitigating the need for standards; thickness measurements by means of MC simulations have been successfully applied

and described in other reports [19–21]. These simulations are used for determining the propagation trajectories of photons and electrons, thus giving information regarding photon counts within the active area of the detector. As a result, the photon counts normalized to one source particle are simulated, dependent on the layer thickness and matrix effects that are present in the investigated multi-layer samples.

The third protocol is based on the fundamental parameter (FP) method. In the literature, the FP method was successfully utilized for quantitative composition determination in several other papers [22–27]. The FP method was applied using the commercially available CrossRoads Scientific XRS-FP software (Version 6.9), which determines the quantitative composition results by means of a standardless characterization procedure. This implies several processing steps, such as powerful noise reduction, peak deconvolution, removal of escape peaks, and a series of artefact mitigating algorithms. As a result, by means of FP equations, the correctly integrated net intensity peaks are converted to the elemental concentration expressed in wt % or mol %. This software is usable for single layer coatings and can be applied to the analysis of elements in the energy range of 5 to 120 keV [28]. It is well known that when using the FP method, the thickness determination accuracy is lower for multi-layer samples, with standard deviations up to ±10%, compared with a single layer sample [29,30]. For improving the measuring accuracy, we used a multi-layer analysis of the thickness and composition (MTF-FP) module integrated in the XRS-FP software, which is based on a standard sample, and can determine the thicknesses with better accuracy, from nanometers to micrometers, for each individual layer. It is important to mention that the combination of a high-resolution collimating system and the application of the FP software permits the detectability of sample contaminants well below 2 wt % [22–27].

## 2.3. Investigated Samples

In the current work, a complex investigation was conducted on different erosion-resistant functional coatings applied as fusion plasma related tiles and lamellae, including alternative multilayers of W and Mo depositions (Figure 2) on different substrates (W, Ti, and FGG). The investigated samples in the current work had the configuration of W on a Mo interlayer with different substrates. The use of the Mo interlayer as a substrate for W was previously reported in the literature, expressing it as an advantage [31]. Also, the Mo interlayer was applied as a buffer for the thermal expansion mismatch between the W layer and the low Z material substrate [31].

The samples were fabricated by means of the combined magnetron sputtering with ion implantation (CMSII) method [32]. The layer deposition methodology behind this technique involves simultaneous magnetron sputtering and high energy ion bombardment of the coating; the magnetron deposition relies on a pulse discharge (U = 40 kV and f = 25 Hz) that generates accelerated positive ions that will be deposited [32–34]. The deposition experiments included synthetized materials such as W and Mo high purity (99.95%) cathodes.

Several aspects have to be mentioned regarding the impurities that the investigated samples could include. In past experiments, we measured different samples expressing impurities in the form of alloy inclusion and dust microparticles on the surface of the samples [3]. The XRF method tolerates the presence of impurities, and the following protocols were applied:

- Corrections of the XRF calibration curves were done for those impurities that were included in the sample alloy. These corrections, expressed as matrix effects, were determined by Monte Carlo simulations. No corrections are needed for low Z inclusions, such as Be or C, because of their transparency to X-rays;
- Impurities as isolated microparticles (dust) on the surface of the samples can easily be avoided because of the use of the polycapillary lens that integrates the measuring area on a small focal spot.

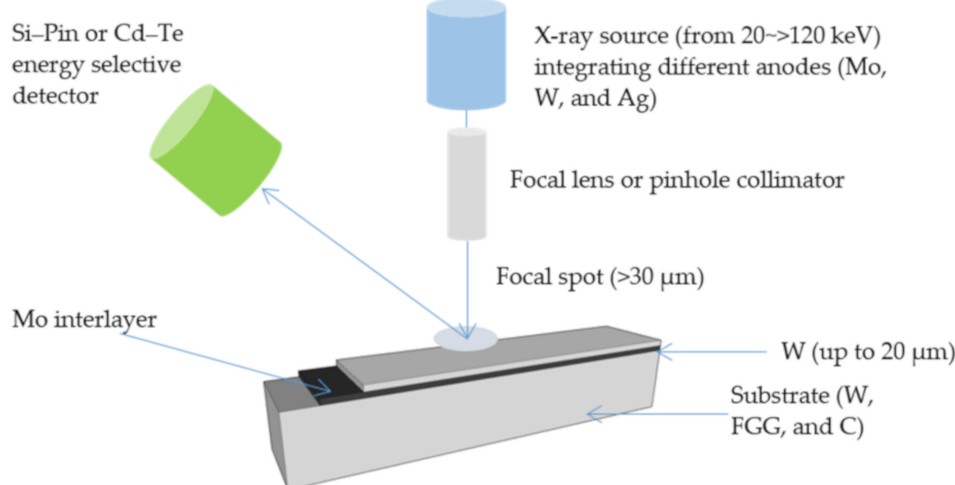

**Figure 2.** XRF experimental principle integrating different components for particular sample configuration; each sample configuration imposed a different approach regarding the experimental geometry.

There is no straightforward approach for conducting a quantitative analysis regarding high Z multi-layer samples. In the literature, there are few studies validating the composition analysis by investigating the K fluorescence lines of high Z elements [30]. Therefore, several customized calibration procedures were applied in order to investigate the K energetic lines of high Z elements. The samples were investigated by means of different XRF methods enhanced by calibration protocols (Table 1).

**Table 1.** Investigated multilayer sample configurations and the applied investigation methods. FGG—fine grain graphite.

| Sample Configuration/Expected Thickness | | Applied XRF Methods | | Calibration Protocols |
|---|---|---|---|---|
| 1. | W (~5.5 μm) and Mo (~6 μm) deposited on W bulk substrates | | | 1. MC simulations and reference samples |
| | | 1. Macro-XRF | | |
| 2. | W (~10 μm) on Mo (~3 μm) on FGG (4 cm) | 2. W K-line XRF | | 2. MC simulations and reference samples |
| 3. | Plasma sputtering eroded multilayer W/Mo on a C bulk substrate | 3. W K-line XRF | | 3. FP and reference samples |

Before conducting XRF measurements, deposition witness samples with thickness known layers of W/Mo on a Ti substrate, named currently as the reference samples, were measured by SEM and the glow discharge optical emission spectroscopy (GDOES) method for calibrating the XRF method. The employed reference samples were previously calibrated by means of SEM and GDOES, and the results are presented in Table 2.

The SEM measurements were carried out by means of an Apreo instrument (Thermo Scientific, Waltham, MA, USA), which produces enlarged images of a variety of specimens, achieving magnification of over 100 000 × (ideal conditions), providing high resolution imaging in a digital format. The SEM images were acquired at a working pressure of 1.5 $\times 10^{-2}$ Pa, a working distance between 8–22 mm, and an electron acceleration voltage of 10–25 kV [35–37].

The GDOES method was applied for in-depth sample analysis. The GDA 750 spectrometer equipment (Spectruma Analytik, Hof, Germany) that was applied presents the advantage of measuring thicknesses of over 100 μm, with an in-depth resolution of 1 nm [34]. Taking into account the high probability aspect of chamber contamination, the investigation of samples with a C-based substrate were avoided.

**Table 2.** Thickness determined for the co-deposited reference samples.

| Sample ID | Mo Layer (µm) | W Layer (µm) |
|:---:|:---:|:---:|
| 1 | 2.3 | Not included |
| 2 | Not included | 4.6 |
| 3 | 2.5 | 4.6 |
| 4 | 4.8 | 5.1 |
| 5 | 2.4 | 9.1 |
| 6 | 2.4 | 13 |
| 7 | 3 | 17.5 |
| 8 | 3.1 | 19.3 |
| 9 | 3.4 | 19.5 |
| 10 | 3.7 | 19.6 |

## 3. Results

In the following sections, both of the XRF methods and protocols were applied according to the stoichiometry of the studied sample and the expected results. The results are expressed as the compositional characterization and quality validation of fusion plasma relevant samples by means of XRF-based methods.

### 3.1. Macro-XRF Analysis Assisted by MC Simulations for W L-Lines Investigation on PFC-Like Lamellae

In the following experiment, the low energy XRF method assisted by MC simulations was applied to PFC relevant lamellae configured with thin W and Mo multi-layers with a W substrate, deposited using the CMSII technique.

The investigation of the PFC lamellae was conducted on a Tomo-Analytic instrument using the macro-XRF method. A low voltage Ag target X-ray excitation source (<50 kV) was used. The spatial resolution of the macro-XRF method was approximated by means of investigating a stainless-steel grid with voids down to 0.4 mm. Therefore, the spatial resolution of the method was determined as ~0.6 mm [12]. As a particularity, the sample holder was built from a PA6 polymer and Al, and it was positioned away from the incident beam, in order to avoid the amplification of the background noise in the fluorescence spectra.

In previous experiments, it was determined that the auto-absorption phenomena of the fluorescence response are present in W layers above 6 µm in thickness [13,14]. Therefore, the macro-XRF method can be successfully applied to the current sample configuration.

Following the measurements, the XRF spectra were determined, thus highlighting the presence of W and Mo corresponding energy peaks (Figure 3).

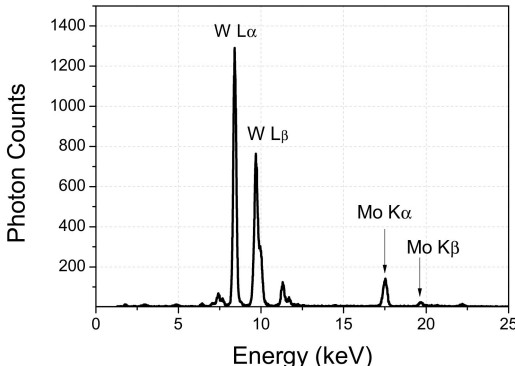

**Figure 3.** XRF spectra of the W/Mo deposited on bulk W (witness sample).

In Figure 3, the Mo signal appeared to be highly attenuated because of the deposited W layer. As a consequence, the Mo/W fluorescence variation in relation to the layer thickness was studied.

　　　The PFC lamellae were fabricated as multi-layers of W and Mo deposited on W substrates. During the lamellae fabrication, deposition witness samples were co-deposited with Mo and W on Ti substrates, and analyzed by means of GDOES and SEM instruments [18]. The GDOES results expressed an atomic concentration decrease of over 50%, observed at a depth value of 5.5 µm for W and at 6 µm for Mo, pointing out the thickness of the deposited layers (Figure 4a). Also, some the oxidation and the presence of C were determinated, although these elements were not detectabe by the XRF methods.

　　　An SEM image of W on the Mo depostion witness sample presents a compact structure, but is slightly irregulated. The W and Mo layers present thickness variations (Figure 4b). The observed agglomerations on the W layer surface could be a result of the polycristaline nature of the deposited material.

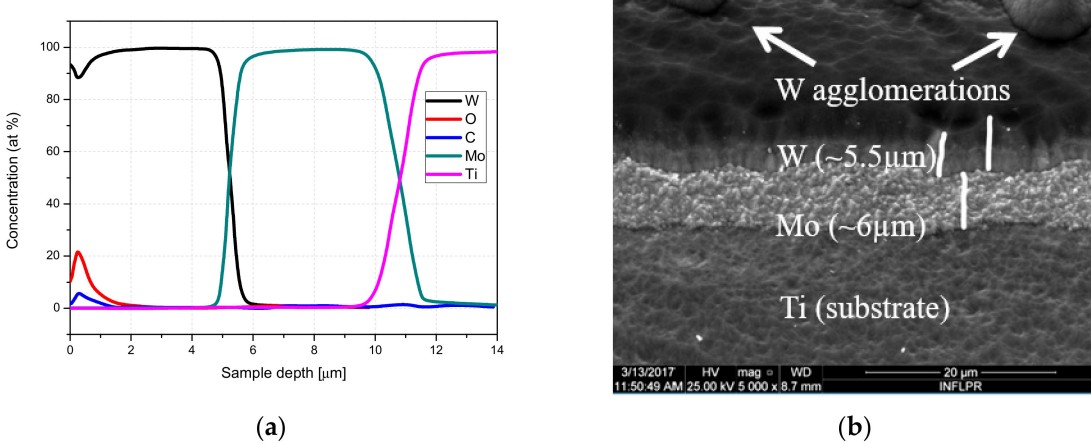

(a)　　　　　　　　　　　　　　　　　　　　　　　　(b)

**Figure 4.** (**a**) In-depth profile of the deposition witness sample; (**b**) cross section SEM image on the deposition witness sample with cluster centers.

　　　The qualitative XRF calibration process implied the energy calibration of the spectrum, establishing the region of interest and the calibration of the integrated areas. From the acquired fluorescence spectra, the Mo and W intensity peaks were analyzed and integrated by means of a Gaussian distribution function. During the calibration process, it was taken into account that the lamella configuration implies the use of W as both a substrate and deposited layer. This could distort the W and Mo fluorescence lines by means of matrix effects. Also, the amplification induced by the substrate on the fluorescence signal has to be well characterized.

　　　Thus, Monte Carlo simulation algorithms were applied in order to study the W substrate's influence on the fluorescence spectra of the deposited layers. Simulations were conducted for calculating the fluorescence signal response for different W layer thicknesses (from 0.5 to 5 µm), with a Mo intermediate layer with a constant thickness value of 6 µm. The substrate influence was simulated for W, Ti, and with no substrate configuration (Figure 5a,b).

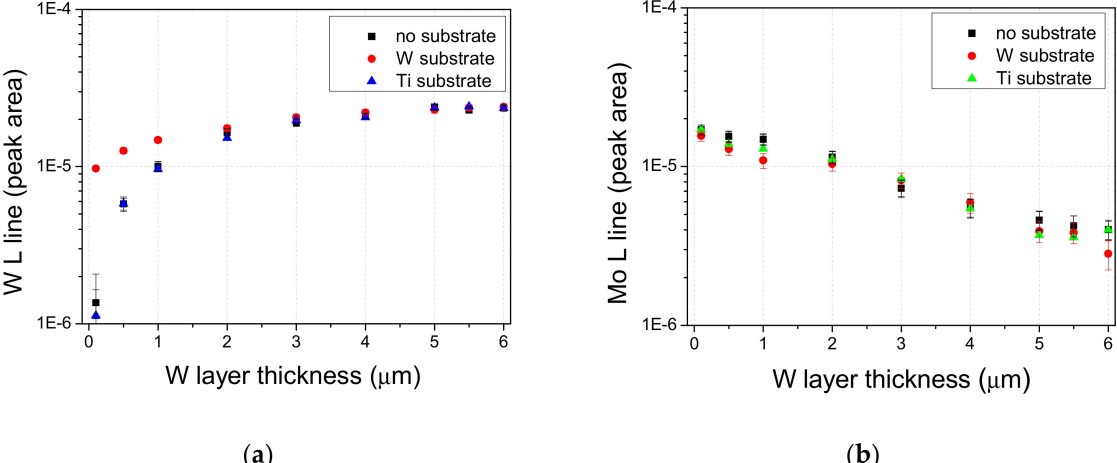

**Figure 5.** Simulated substrate dependence of the L-line fluorescence intensities for (**a**) W and (**b**) Mo.

The simulated curve determined for the configuration without a substrate was applied as reference, and it was noticed that the induced amplification of the W substrate increased when the W layer was thinner that 3 μm (Figure 5a). As expected, the presence of Ti and W substrates did not distort the fluorescence signal response for the Mo L energetic lines (Figure 5b).

The validation of the simulated curves and the determination of a correction factor between the experimental and simulated data were conducted. The correction factor was determined by comparing the experimental and simulated Mo/W ratio for a bulk Mo and W sample configuration. The determined correction factor was applied to the Mo/W simulated curve in relation to the W layer (Figure 6a). The experimental data point overlaps the simulated calibration curve in the thickness region relevant to the non-eroded PFC lamellae.

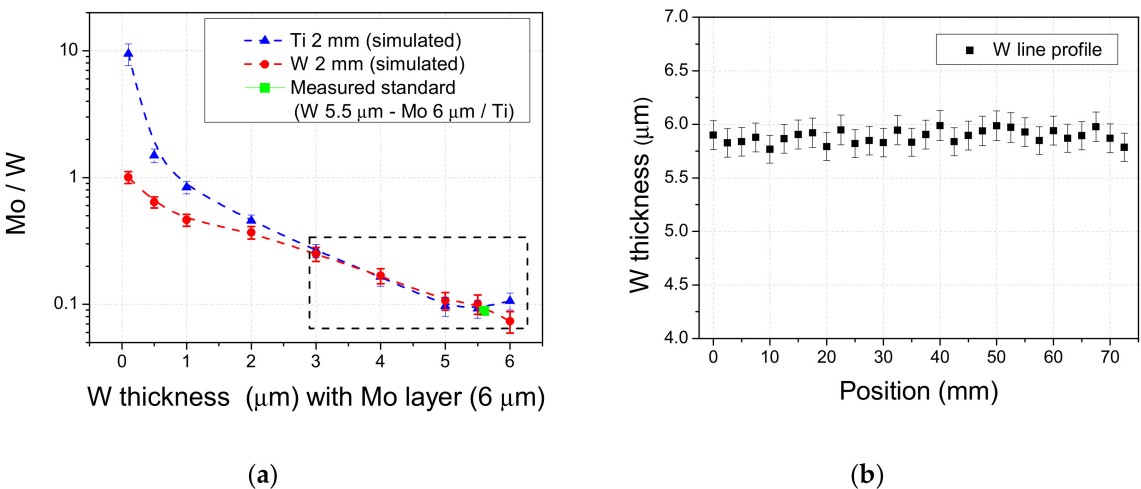

**Figure 6.** (**a**) Validated calibration curve by means of a correction factor; the green data point represents the experimentally determined mean Mo/W ratio from integrating multiple points on the standard sample (W is 5.5 μm and Mo is 6 μm). (**b**) Macro-XRF determined W profile on plasma facing component (PFC) relevant lamella.

The substrate influence is insignificant for the thick deposited layers of W that are present in the lamellae configuration. Therefore, the fitting equation determined from the simulated calibration curve was further applied.

The W profile on the deposited lamella was determined by means of a calibration curve with the previously applied correction factors (Figure 6b). The variations in the thickness profile were

expected because of the surface agglomerations of W that were observed in the SEM image (Figure 4 b). Thus, it was proven that the micro-XRF assisted with MC simulations could determine stoichiometry variations for thin W depositions, with a sensibility of up to 10% in relation to the thickness values.

*3.2. Tile Validation as a PFC-Like Candidate by Means of the W K-Line XRF Method Assisted with MC-Based Simulations*

In this study, a tile deposited by means of the CMSII method was investigated, in order to provide information regarding the deposition uniformity and the presence of deposition anomalies caused by possible shadowing effects. Measurements were conducted for validating the tile as a possible PFC candidate, which could be integrated in the fusion reactor chamber.

The reference samples and FP- and MC-based calibration protocols were applied for investigating the PFC tile. The deposition uniformity was studied by non-destructive XRF methods applied on PFC samples that are intended to be integrated in the tokamak inner-wall. Thus, the W K-Line XRF method was applied for investigating a large area (326 mm × 29.6 mm) tile, co-deposited by means of CMSII method in a configuration of W on Mo with FGG substrate (4 cm) [38]. In order to apply MC simulations, the FGG substrate was considered equivalent to C, having a slightly lower density of 1.3 g/cm$^3$.

The FGG substrate had a 4 cm thickness and the sample was mounted on a specially designed holder, maintaining it perpendicularly to the primary X-ray beam. The W K-line XRF spectra comprised a large energetic range, thus providing information regarding W K- and L-lines and Mo K-lines (Figure 7a). An overview of the FP spectra processing steps, such as background noise reduction and peak integration by means of Gauss fitting, are further highlighted in Figure 7b.

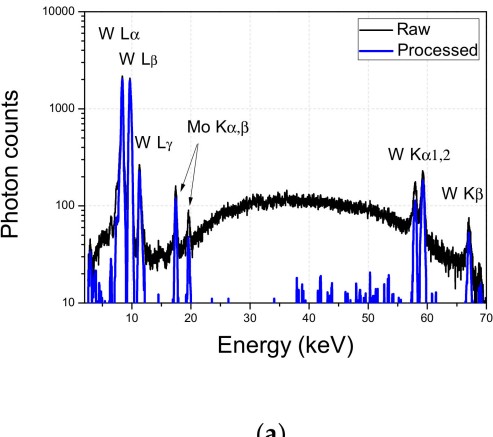 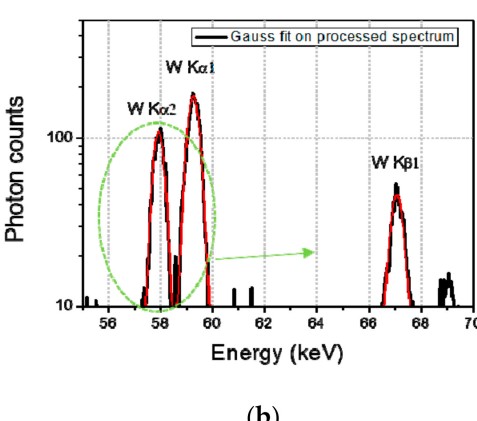

(**a**)                                                                 (**b**)

**Figure 7.** XRF spectra of the measured tile: (**a**) W K-line XRF raw spectrum with an overview of the W characteristic lines and highly amplified background noise due to the FGG substrate; Background mitigation on processed spectrum by means of FP parameter software; (**b**) Gauss functions integrations applied on W K-lines.

In the current experimental configuration, the X-ray beam is collimated in order to obtain a focal spot with a diameter of 0.8 mm on the surface of the sample. Prior to the investigation of the PFC tile, an energetic calibration of the detector was conducted based on the Americium-241 element with a well-defined spectrum, thus converting the detector channels to photon energy (keV).

The currently applied standards, as reference samples, were designed in a W and Mo multi-layer configuration on a Ti substrate, and were deposited by means of the CMSII technique (Table 2). The reference samples were an in-depth profile calibrated trough the GDOES. Also, SEM measurements on the crater-like erosions determined from the GDOES analysis were applied to validate the determined results on the reference samples [32–34].

The GDOES calibrated reference samples (Table 2) were further applied as standards in the determination of the thickness calibration curve. For this study, the reference samples were suitable for

calibrating, considering the fact that the investigated tile is foreseen to have a W thickness around 10 μm. However, it is necessary to determine whether the Ti substrate found in the reference samples can be overlooked as a matrix error producing a factor.

Analyzing the standard samples by means of W K-line XRF, the conversion of the integrated areas of the spectral lines into the layer thickness for Mo and W elements was conducted. A quasi-linear relation was observed between the thickness of the W layer and the integrated peak area of the K-energetic lines (Figure 8a), resulting in a non-saturated fluorescence signal for the region of interest. Furthermore, the Mo calibration curve needs to be determined in order to conduct a full characterization of the co-deposited tile sample. We considered that the tile has a Mo deposition as the interlayer between the substrate and the W layer. Thus, the calibration curve expresses the integrated peak area of the Mo $K_{\alpha,\beta}$ energetic lines in relation to the thickness of the W layer (Figure 8b).

Additionally, MC simulations were carried out on the Mo fluorescence signal response (Figure 8b), thus predicting the response of the normalized peak intensity of Mo in relation to different W deposition thicknesses. In order to make a comparison between the experimental data, simulations were conducted to include the configurations of the reference samples. The total simulated configurations included W layers with thickness from 250 to 40 μm, deposited on a Mo layer with a fixed value of 2.4 μm, and a Ti substrate (2 mm). A good agreement between the simulated and measured data is observed in Figure 8b.

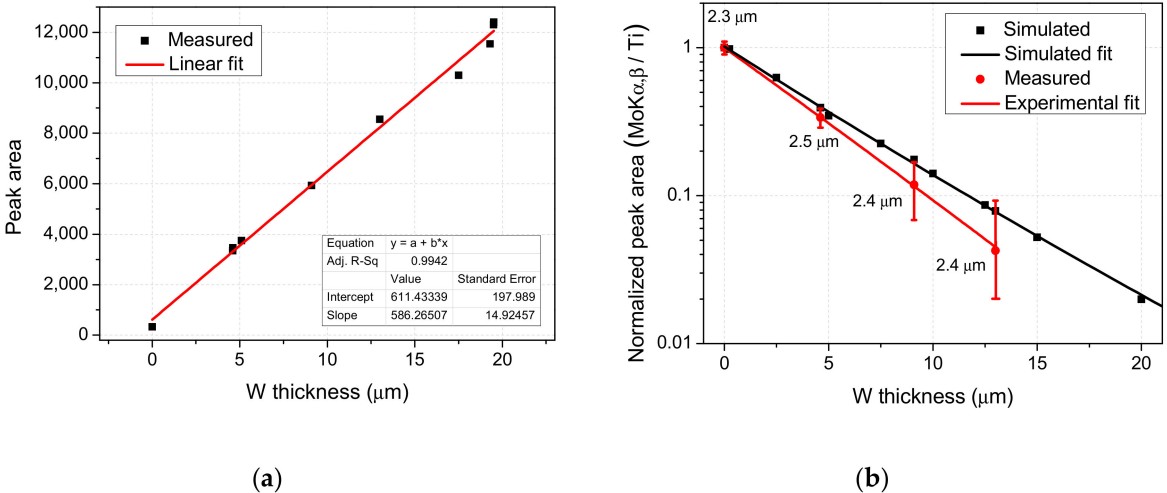

(**a**)                                                                                                        (**b**)

**Figure 8.** Calibration curves: (**a**) W calibration curve that expresses the normalized peak areas of W $K_{\alpha,\beta}$ in relation to the W layer thickness; (**b**) overlapping of experimental and simulated calibration curve expressing the intensity response of Mo $K_{\alpha,\beta}$ peaks for a quasi-constant Mo thickness layers (2.4 ± 0.1 μm) in relation to the W layer thickness.

By means of the W K-line XRF method, a profile analysis of the full sample length (325 mm) was conducted (Figure 9), with a 5-mm measuring step and with an integrating time of 60 s per measuring point. The error bars were calculated as the relative standard deviation, representing the square root of the integrated peak area.

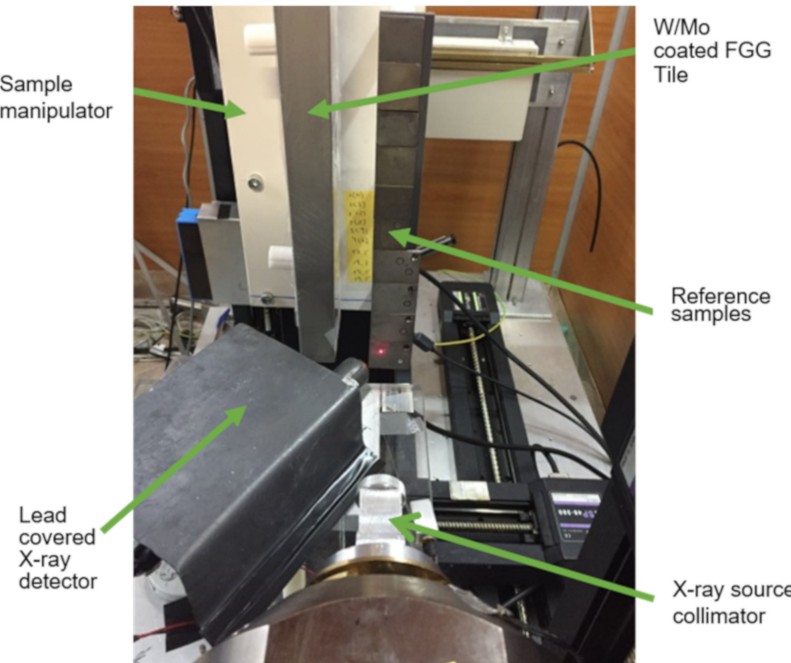

**Figure 9.** W K-line XRF experimental configuration with high energy X-ray source and Cd–Te diode-based energy selective detector.

Before correcting the measurements based on the determined calibration curves for Mo and W, certain MC simulations were applied for calculating the influence of the substrate and sample holder on the W K-lines. The amplification factor of the intensity lines was simulated for a W layer with a tile relevant thickness of 10 µm (Figure 10a), in relation to the element thicknesses applied as the substrates (Ti, C, and Si).

Compared with the no-substrate sample configuration that was simulated as the reference, the mean amplification factor determined by the Ti substrate on the standard samples with a thickness of 2 mm was 17%. Meanwhile, it was observed that the amplification factor for the thick (4 cm) C substrate found in the tile can determine a W signal amplification up to 27.5% (Figure 10a).

The measured line profiles were corrected by compensating the W fluorescence amplification, thus resulting in the calibrated thickness line profiles of W and Mo on the full length of the tile sample (Figure 10b).

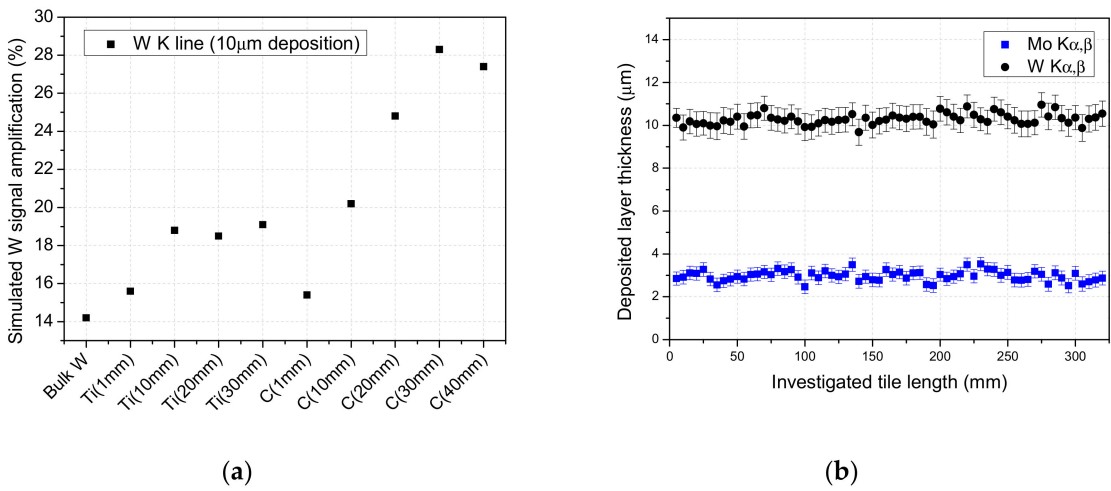

(**a**)                                        (**b**)

**Figure 10.** (**a**) Dependency of the amplification factor of the W photon counts on the substrate composition and thickness. (**b**) Amplification factor corrected and calibrated thickness profiles of Mo and W layers determined on the PFC tile.

The calibrated line profiles on the investigated PFC tile (Figure 10b) were determined based on protocols relying on the reference samples and MC simulations. The W K-line XRF method provided information in a non-destructive manner regarding the deposition thickness uniformity for both Mo and W elements deposited as multilayers on the FGG substrate. Quasi-uniform variations with no anomalies were noticed for both Mo and W along the investigated profile length, thus validating the tile as a proper PFC candidate.

### 3.2.1. W K-Line XRF-MTF-FP Measurements for Erosion Study of Plasma Eroded Functional Coatings

The following W K-line XRF measurements were carried out in order to investigate the erosion pattern on a plasma exposed tile consisting of a multilayer of W and Mo on a C substrate. As the total thickness of W in the sample is above the W L-line saturation threshold, in this experiment, the W K-lines were investigated. The proper protocol applied to these measurements is included as MTF-FP module in the XRS-FP software.

Using this module for the K-line XRF of high Z materials like W, numerous advantages are available compared with the L-line analysis conducted by means of classical XRF methods. In order to output the most reliable quantitative results, the MTF-FP module requires some calibrations based on standard samples. For this reason, the reference samples described in Table 1 were used in the MTF-FP module calibration. The reference samples spectra were processed by means of MTF-FP; the W K- and L-lines are visible in the processed spectra (Figure 11).

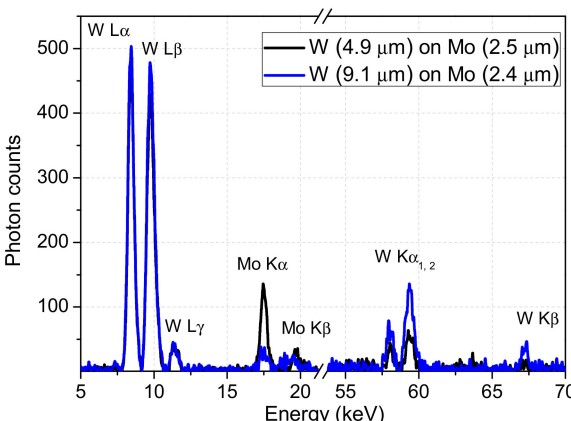

**Figure 11.** MRS-FP corrected spectra of typical characteristic W K- and L-lines from standard samples with the following configurations: W (19.5 μm)/ Mo (2.5 μm) and W (4.9 μm)/Mo (2.5 μm).

By means of the MTF-FP module, the following calibration curves were determined for W and Mo (Figure 12). The measured intensities' responses in relation to the theoretical intensities are expressed by the theoretical correction coefficients (TCC). The predicable linear fit of the determined TCC that can be observed in Figure 12 validated the fact that the MTF-FP-based calibration protocol can be successfully applied, thus enhancing the W K-line XRF method to be applied on relatively thick W-layer containing samples.

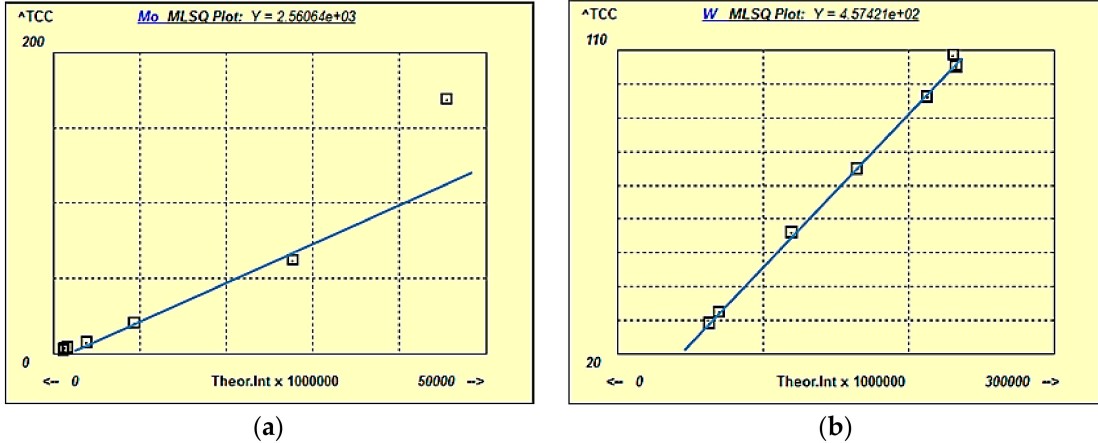

**Figure 12.** Theoretical correction coefficients (TCC) calibration curves: (**a**) for W and (**b**) Mo elements were abscise is expressed as theoretical intensities and ordinate as measured intensities.

Following the results above, we applied the FP enhanced W K-line XRF method for the quantitative mapping of the plasma sputtered W/Mo surface. The mapping was performed on 32 points × 34 points, on an area of 80 mm × 85 mm, with a 2.5 mm measuring step (Figure 13a).

XRF line profiles on the eroded sample shows saturated profile of W L-line compared with the W K-line profiles. As expected, it can be observed in the arbitrary units expressed spectra (Figure 13b,c) that the W L-line is saturated because of the coating thickness; the erosion pattern of the Mo K-line does not rigorously follow the K- or L-line of W, which could identify some areas of the exposed Mo layer. A gradual erosion of W was observed while exposing the Mo layer (Figure 13d,e).

Relying on the determined TCC, further data processing was conducted in order to determine the quantitative results expressed in the layer thicknesses for both the Mo and W layers. Therefore, an erosion pattern was determined on the exposed sample.

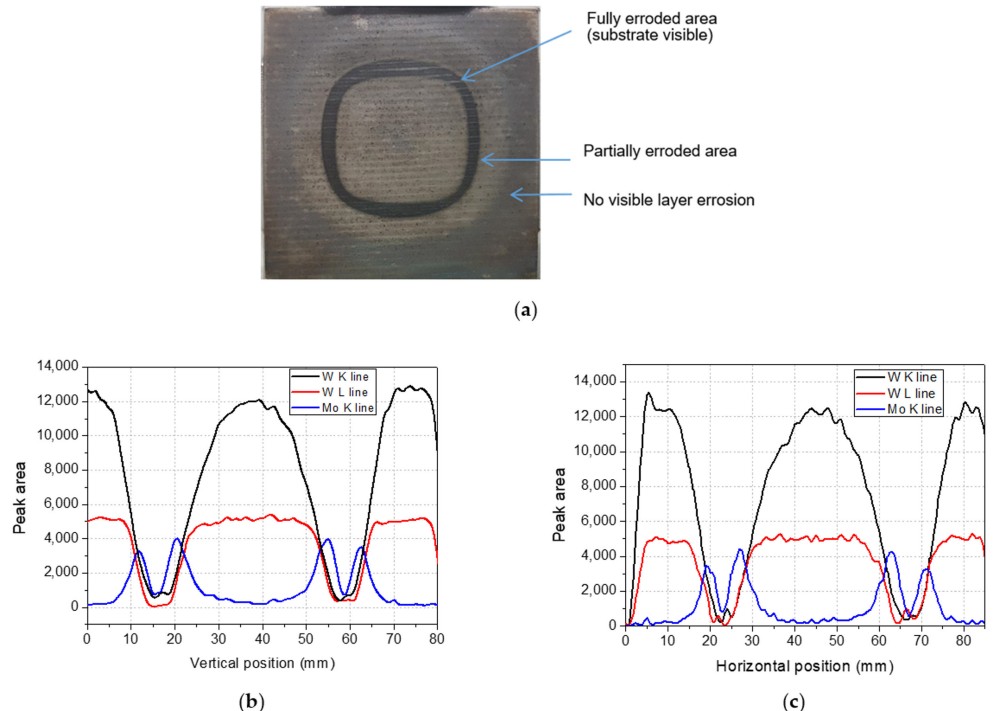

**Figure 13.** *Cont.*

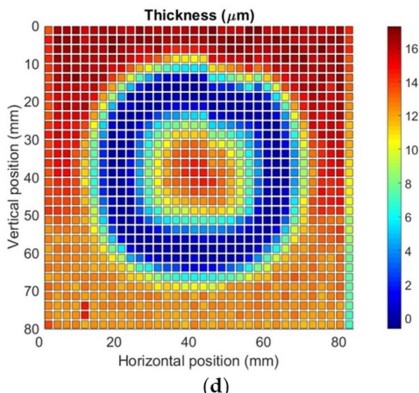 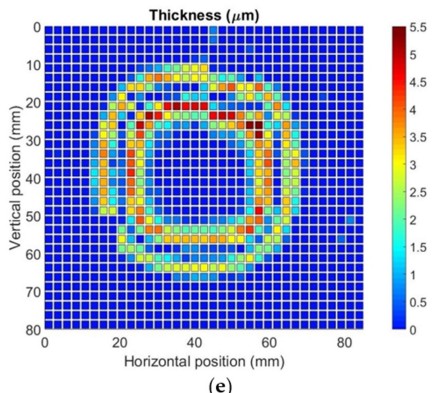

(**d**) (**e**)

**Figure 13.** (**a**) Eroded sample photo; (**b**) 2D vertical XRF profile; (**c**) 2D horizontal XRF profile; XRF mapping of plasma sputtered W/Mo on C substrate sample by MTF-FP applied to W/Mo K-line XRF data: (**d**) 3D W thickness mapping and (**e**) 3D Mo thickness mappings.

The partial and total erosion of the W and Mo layers exposed to magnetron plasma sputtering were successfully determined by means of the W K-line XRF method enhanced by the MTF-FP protocol.

## 4. Conclusions

In our laboratory, we successfully enhanced and applied X-ray fluorescence-based methods for the adequate and non-destructive analysis of erosion resistant functional coatings. The feasibility of the investigation tools was proven; thus, the enhanced methods can provide a fast analysis with millimeter range spatial resolution on relevant layers, designed to be applied in extreme conditions.

X-ray fluorescence assisted by fundamental parameter dedicated software was proven to be a powerful tool that can determine results expressed in concentrations and layer thickness. Because of the automatization provided by the motorized translation axis, the XRF-based methods were validated as being well suited for deposition mappings conducted on samples in single- or multi-layer configurations. By integrating well defined protocols, the processing of raw fluorescence spectra into quantitative and qualitative results was successfully realized and determined. XRF-based methods were successfully applied as non-destructive techniques on fusion relevant samples to be applicable in extreme conditions, providing an alternative to other well-known inspection methods. Thus, thickness determination was realized for the tile and lamellae obtained by the combined magnetron sputtering with ion implantation method. Additional experiments are currently being planned to compare the before and after fusion plasma exposed samples by means of enhanced X-ray fluorescence methods, in order to make further contributions to understating the plasma determined erosion and deposition phenomena.

An advantage of the methods qualified in this study is their ability to cope with samples containing impurities in the form of alloys as minor components and/or contaminants, or dust microparticles on the surface of the samples.

**Author Contributions:** Conceptualization, I.T.; methodology, C.D., M.L., and I.T.; investigation, M.L. and C.D.; writing (original draft preparation), M.L. and I.T.; writing (review and editing), C.D. and M.L.; supervision, I.T.

**Funding:** This work was financed by the Ministry of Research and Innovation in the frame of the Nucleus Program (contract 18 13 01 01), "Applicative research with lasers, plasma, and radiations for the development of new and emergent technologies, materials, and advanced devices".

**Acknowledgments:** M.L., C.D., I.T. acknowledge the Romanian Ministry of Research and Innovation through the Core Program PN 19 15 01 01 (contract no. 16N/2019) for the financial support.

**Conflicts of Interest:** The authors declare no conflict of interest.

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
