# Peer review of "Enhanced XRF Methods for Investigating the Erosion-Resistant Functional Coatings"

_coatings, doi:10.3390/coatings9120847_

Round 1

Reviewer 1 Report

The manuscript is devoted to the improvement of XRF method and the development of a methodology for spectral data analysis to study the coatings used for plasma facing components. The results are novel and have some interest for potential applications. However, some comment should be addressed to the authors before the manuscript can be accepted.

The authors should check the manuscript, as there are some errors in the text (for example, in captions of Fig. 1, 2 and in other places). The manuscript is mainly devoted to the XRF method, and unfortunately, there is no information about the deposited coatings. Nothing are written about coating deposition methodology. What was the reference sample? In addition, some SEM or GDOES data should be given to readers so that they understand what was the microstructure, thickness and composition uniformity of the deposited coatings. At least some of this data. What was the purity of targets used for magnetron sputtering? Is it possible to use your protocols (methodology) to analyze the layers thicknesses in the presence of coating’s impurities? The article does not contain Conclusions or the authors wrote it in the Discussion section.

Author Response

Reviewer 1:

The methodology for spectral data the manuscript is devoted to the improvement of XRF method and the development of an analysis to study the coatings used for plasma facing components. The results are novel and have some interest for potential applications. However, some comment should be addressed to the authors before the manuscript can be accepted.

The authors should check the manuscript, as there are some errors in the text (for example, in captions of Fig. 1, 2 and in other places).

Captions for figure 1 and 2 were corrected in order to avoid any confusion as:

For Figure 1 - Calibration protocol for: standards measurements, Monte Carlo simulations and fundamental parameters (FP) approach;

For Figure 2 - XRF experimental principle integrating different components for particular sample configuration;

The manuscript is mainly devoted to the XRF method, and unfortunately, there is no information about the deposited coatings.

Information regarding the deposited coatings was added in the manuscript expressed in the following paragraph:

Investigated samples in the current work had the configuration of W on Mo interlayer with different substrates. The use of Mo interlayer as a substrate for W was prior reported in the literature expressing an advantage [31]. Also, Mo interlayer was applied as a buffer for the thermal expansion mismatch between the W layer and the low Z material substrate [31].

[31] Tamura S. et al N 2003 J Nucl. Mater. 313-316 250; and Jihong D. et al. 2005 Surf. Coat. Technol. 198 169;

Nothing is written about coating deposition methodology.

Additional information was provided regarding the applied deposition technique in the following paragraph:

In the current work, samples were fabricated by means of Combined Magnetron Sputtering with Ion Implantation (CMSII) method [32]. The layer deposition methodology behind this technique involve simultaneous magnetron sputtering and high energy ion bombardment of the coating; magnetron deposition relies on a pulse discharge (U=40kV, f=25 Hz) that generates accelerated positive ions that will be deposited [32-34].

[32] C. Ruset, et al., Phys. Scr. T128 (2007) 171-174, doi:10.1088/0031-8949/2007/T128/033;

[33] C. Ruset, et al., Fusion Engineering and Design 86 (2011) 1677-1680;

[34] C. Ruset, et al., Fusion Engineering and Design 84 (2009) 1662-1665.

What was the reference sample?

The reference samples are deposition campaign witness probes that are destructively tested by means of SEM and GDOES instruments for thickness evaluation. Now, this information is provided in the manuscript:

“deposition witness samples with known thickness layers of W/Mo on Ti substrate, named currently as reference samples, were measured by SEM and GDOES for calibrating the XRF method”.

In addition, some SEM or GDOES data should be given to readers so that they understand what was the microstructure, thickness and composition uniformity of the deposited coatings. At least some of this data.

In this work, the exposure to destructive analysis methods of CMSII fabricated tiles and lamellae was avoided.

Nevertheless, GDOES and SEM measurements results for reference sample (deposition witness sample) were conducted and inserted in the manuscript (Figure 4 a, b). The following paragraphs and figures were inserted in the results section:

“During the lamellae fabrication, deposition witness samples were co-deposited with Mo and W on Ti substrate and analyzed by means of GDOES and SEM instruments [18]. The GDOES results expressed an atomic concentration decrease over 50%, observed at a depth value of 5.5 µm for W and at 6 µm for Mo, pointing out the thickness of the deposited layers (new figure - figure 4. a). Also, some oxidation and the presence of C were determined, although these elements are not detectable by the XRF methods.

SEM image of W on Mo deposition witness sample presents a compact structure, but slightly irregular. The W and Mo layers present thickness variations (new figure - figure 4. b). The observed agglomerations on the W layer surface could be a result of the polycrystalline nature of the deposited material.”

[18] Lungu, M; Tiseanu, I.; et al, Preparation and analysis of functional fusion technology related materials, Romanian Journal of Physics, 60(3-4):560-572, (2015);

During the deposition of each sample by means of what was the purity of targets used for magnetron sputtering?

The following information was introduced in the investigated samples subsection:

“The deposition experiments integrated synthetized materials such as: W and Mo high purity (99.95 %) cathodes.”

Is it possible to use your protocols (methodology) to analyze the layers thicknesses in the presence of coating’s impurities?

We consider that this question treats an important aspect regarding the impurities often found in the relevant fusion coatings. Therefore, we included the following answer in the manuscript: 

In past experiments, we measured different samples containing impurities in the form of alloy minor components and/or contaminants or dust microparticles on the samples surface [3]. The XRF method is able to tolerate the presence of impurities and following protocols were applied:

For minor components of alloys the thickness calibration curves could be corrected by Monte Carlo simulations of the matrix effects induced in the standard samples by the presence of the impurities. Impurities as isolated microparticles (dust) on the samples surface can easily be avoided due to the use of polycapillary lens that integrates the measuring area on a small focal spot. For typical deposited contaminants as Be and C there is no need to apply significant corrections due to their transparency to X-rays;

 [3] Tiseanu, I. et al. Surface and Coatings Technology, Vol 205, supplement 2, Pages s192-s197, 2011, doi.org/10.1016/j.surfcoat.2011.03.049;

The article does not contain Conclusions or the authors wrote it in the Discussion section.

Section 4 was renamed as conclusions.

Reviewer 2 Report

In this paper, X-ray fluorescence-based methods were used for the analysis of erosion-resistant functional coatings. It is an interesting work, however, it needs more effort in the representation of results and discussions. Language should be revised carefully. This reviewer recommends the publication of the present manuscript in
Coatings. However, the following modifications should be addressed by the authors:
1. The abstract should provide some of the key results.
2. In the introduction section: Some pioneering and recent works applying XRF for the determination of elements should be discussed in detail.
3. Section 3.1, authors should represent results and discussion instead of the experiment details. Any detail should be represented in 2.
4. Please check and modify the language of the entire manuscript.
5. Section 3. should be "Results" since the authors suggested another section for "Discussions".
6. Overall, the entire manuscript should be well organized and represented.

Author Response

Reviewer 2:

In this paper, X-ray fluorescence-based methods were used for the analysis of erosion-resistant functional coatings. It is an interesting work; however, it needs more effort in the representation of results and discussions. Language should be revised carefully. This reviewer recommends the publication of the present manuscript in Coatings. However, the following modifications should be addressed by the authors:

The abstract should provide some of the key results.

The following paragraph containing key results was inserted in the abstract:

“The deposition uniformity was successfully investigated for different PFC relevant tiles and lamella shaped samples with W layers below and over the W L Line saturation thickness. Also, the 2D thickness mapping capability of the XRF method was demonstrated by studying the plasma post-erosion pattern.”

In the introduction section: Some pioneering and recent works applying XRF for the determination of elements should be discussed in detail.

The following paragraph containing details regarding the XRF method was introduces in the introduction section:

“More than that, the XRF method is considered an important tool with the applicability in domains such as environment [5], food [6], metallurgy [7], etc. In general, the key advantages of XRF based methods are expressed as: fast measuring, requires no consumables and it is highly versatile [5].”

[5] E. Marguí, I. Queralt, M. Hidalgo, Determination of cadmium at ultratrace levels in environmental water samples by means of total reflection X-ray spectrometry after dispersive liquid-liquid microextraction, Journal of Analytical Atomic Spectrometry 28 (2013) 266–273;

[6] A. Otaka, A. Hokura, I. Nakai, Determination of trace elements in soybean by X-ray fluorescence analysis and its application to identification of their production areas, Food Chemistry 147 (2014) 318–326;

[7] D. Šatović, V. Desnica, S. Fazinić, Use of portable X-ray fluorescence instrument for bulk alloy analysis on low corroded indoor bronzes, Spectrochimica Acta, Part B 89 (2013) 7–13;

Section 3.1, authors should represent results and discussion instead of the experiment details. Any detail should be represented in 2.

General information regarding the experimental configuration were removed from the results and discussion sections and inserted in the methods description section. However, to have a better understating regarding the results, some experimental details were considered as necessary to be maintained in the results dedicated section (e.g. sample holder that could influence further the detection quality, the results determined with a resolution dependent on the type of used X-ray focusing lens);

Please check and modify the language of the entire manuscript.

The language of the entire manuscript was checked, modified and highlighted with track changes.

Section 3. Should be "Results" since the authors suggested another section for "Discussions".

In order to avoid any confusion, Section 3 was renamed as “Results”.

Overall, the entire manuscript should be well organized and represented.

The entire manuscript was supposed to an extensive revision in order to enhance its structure, especially in the result section divided in 3 major subsections expressing the applied protocols for the imposed different configuration samples. Also, the “results and discussion” section was renamed as “results” and the “discussion” section was renamed as “conclusions”.

Round 2

Reviewer 1 Report

The authors checked and revised the manuscript according to Reviewer's comments. Some grammatical errors still remained, however, they do not affect the quality of the manuscript.Therefore, the manuscript can be accepted in the current form.

Reviewer 2 Report

The manuscript can be accepted for publication in its present form.